# Multi-chiral materials comprising metallosupramolecular and covalent helical polymers containing five axial motifs within a helix

Francisco Rey- Tarrío[1], Emilio Quiñoá [1], Gustavo Fernández [2] ✉ & Félix Freire [1] ✉

Supramolecular and covalent polymers share multiple structural effects such as communication mechanisms among monomer repeating units, which are related to their axial helical structure. Herein, a unique multi-helical material combining information from both metallosupramolecular and covalent helical polymers is presented. In this system, the helical structure described by the poly(acetylene) (PA) backbone (*cis-cisoidal*, *cis-transoidal*) guides the pendant groups in a fashion where a tilting degree emerges between a pendant and the adjacent ones. As a result, a multi-chiral material is formed comprising four or five axial motifs when the polyene skeleton adopts either a *cis-transoidal* or *cis-cisoidal* configuration: the two coaxial helices—internal and external—and the two or three chiral axial motifs described by the bispyridyldichlorido Pt[II] complex array. These results show that complex multi-chiral materials can be obtained by polymerizing appropriate monomers that combine both point chirality and the ability to generate chiral supramolecular assemblies.

In recent years, covalent and supramolecular helical polymers have attracted the attention of the scientific community to replicate the structure/function relation found in helical biomacromolecules such as peptides, proteins, DNA or polysaccharides[1–12]. Synthetic helical polymers, including both covalent and supramolecular, are archetypal systems exhibiting unique helical structures with various functionalities that may even exceed those found in nature. As a result, stimuli-responsive materials can be obtained, making possible to control the *P* (right-handed)/*M* (left-handed) screw sense and/or elongation in covalent polymers[13–18] and molecular packing (*P/M*, J-type, H-type, and so on) by the presence of different external stimuli (e.g., solvent, pH, temperature, metal ions, chiral additives, or light)[19–38]. In addition, in both supramolecular and covalent helical polymers, structural effects such as Sergeants and Soldiers Effect[39–43], Majority Rules[44–46], Chiral Coalition[47], Chiral Accord[48], or Chiral Conflict[49,50] occur when a chiral

and an achiral monomer—or two chiral monomers or building blocks—are copolymerized. This fact indicates that the parameters that govern the communication mechanism between monomers along the supramolecular or covalent helix are the same. This has allowed the development of chiral materials that combine these two structural motifs, where supramolecular arrays of dendronized planar molecules or azo groups are found at the periphery of the covalent helix, leading to its stabilization[51–55]. Interestingly, in literature there are also other examples where a chiral supramolecular helical array of the pendant groups is produced, although the polymer main chain is not folded into a preferred screw sense helix[56–60].

Recently, our group reported a poly[oligo(*p*-phenyleneethynylene)acetylene] (POPEPA) in which the chiral information of the pendant group is harvested by the polyene backbone (*P/M* helix) through the *P/M* axial arrangement of the oligo(*p*-

[1]Research Center in Biological Chemistry and Molecular Materials (CiQUS), Universidade de Santiago de Compostela, E-15782 Santiago de Compostela, Spain. [2]University of Münster, Institute of Organic Chemistry, Corrensstraße 36, 48149 Münster, Germany. ✉e-mail: fernandg@uni-muenster.de; felix.freire@usc.es

phenyleneethynylene) (OPE) groups used as spacers[61]. As a result, a POPEPA can adopt a multi-helical scaffold where four different axial motifs coexist in the helical material: the two coaxial helices—internal (helix 1) and external (helix 2)—and the two helices described by the OPE axial array (helices 3 and 4)[61].

Herein, using rational monomer design, we have achieved unique multi-helical materials that exist in up to five helical conformations. To this end, we selected a metal-containing monomer, consisting of an asymmetric bispyridyldichlorido Pt(II) complex, functionalized on one edge with a polymerizable terminal alkyne and with a chiral group on the other edge to induce a screw sense in the helical polymer. The choice of the Pt(II) center is biased by the versatile self-assembly behavior and interesting photophysical properties of metal-based monomers[62–66]. Particularly, bispyridyldichlorido Pt(II) complexes with an extended aromatic surface are privileged scaffolds to prepare a large variety of supramolecular structures by different aggregation pathways and multiple weak non-covalent interactions[67–72]. Hence, an asymmetric bispyridyldichlorido Pt(II) complex should meet all the requirements to generate a multi-helical scaffold.

## Results

To validate our hypothesis, two different pyridine ligands, one containing either a chiral L-valine or an L-alanine methyl ester residue, and another one bearing a 4-alkynyl group were synthesized and complexed with a Pt(MeCN)$_2$Cl$_2$ salt to yield the corresponding unsymmetrical bispyridyldichlorido Pt(II) complexes (mono-(S)-1 and mono-(S)-2 respectively; see the Supplementary Information for synthesis and characterization of the complexes; Supplementary Figs. 1–8 and Supplementary Tables 1–4). The terminal alkyne group enables the

further polymerization of mono-(S)-1 and mono-(S)−2 using [Rh(nbd)Cl]$_2$ as catalyst (nbd = 2,5-norbornadiene) to obtain poly-(S)−1 and poly-(S)−2, in good yield and high content of cis-double bonds (Fig. 1).

UV–vis studies of mono-(S)-1 and mono-(S)-2 show spectra with four bands between 350 and 250 nm (vide infra). UV–vis theoretical calculations for mono-(S)-1 (time-dependent density functional theory) (TD-DFT) together with either rCAM-B3LYP and wB97XD density functional and LANL2DZ basis set [TD-DFT(wB97XD)/LANL2DZ], TD-DFT(rCAM-B3LYP)/LANL2DZ)], were performed on the molecular structure obtained after DFT optimization (see Supplementary Figs. 18, 19). Both theoretical calculations show a good match between the experimental and the calculated ECD trace, being slightly better when the functional wB97XD is used. From the TD-DFT (wB97XD)/ LANL2DZ theoretical studies, we identified which fragments of mono-(S)-1 are responsible for these absorption bands. Thus, the UV-band at higher wavelength—ca. 345 nm—corresponds to MLCT involving the core of the bispyridyldichlorido Pt(II) complexes, the band at ca. 305 nm corresponds to the para-ethynylpyridyl moiety, while the band at 280 nm is attributed to the para-pyridylbenzamide of (L)-valine-methyl ester complexed to the Pt(II) ion. Finally, the band at 250 nm is assigned to the benzamide group of the chiral moiety (See Fig. 1c and Supplementary Figs. 18, 19).

Intriguingly, UV–vis studies of poly-(S)-1 and poly-(S)-2 show an interesting phenomenon. Thus, while in moderately polar solvents such as THF or DCM, poly-(S)-1 adopts a stretched helix (cis-transoidal), in more polar solvents such as DMF, the polymer adopts a compressed scaffold (cis-cisoidal). This change in elongation is explained by the hypsochromic shift experienced by the polyene band (from 417 to 380 nm) when comparing a DCM/THF solution to a DMF

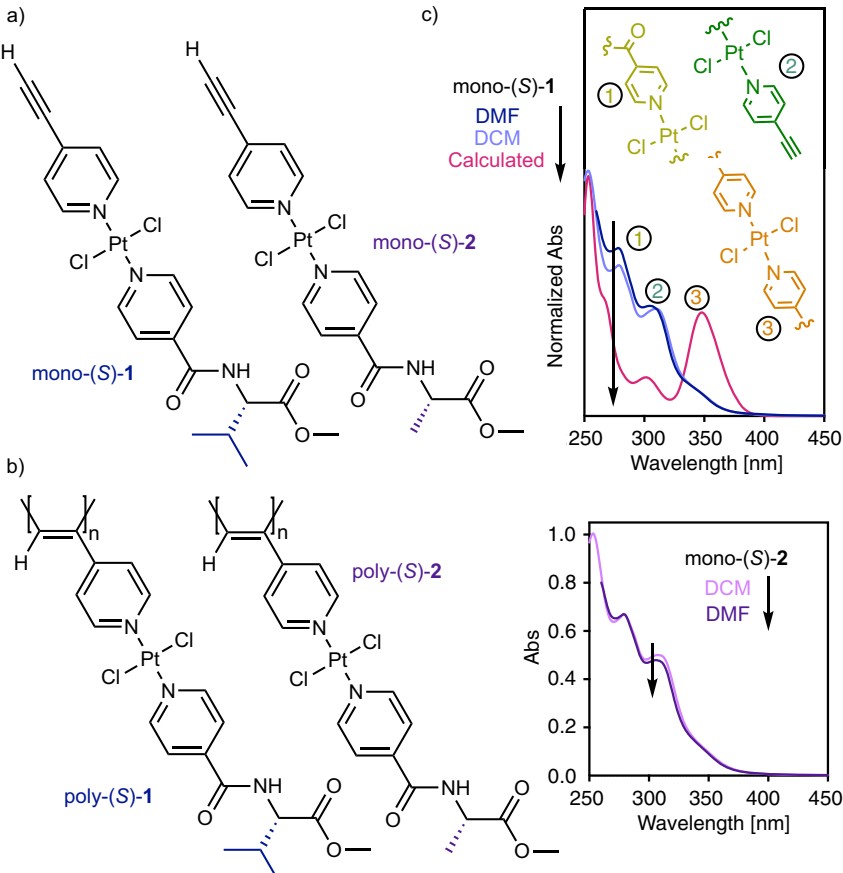

**Fig. 1 | Monomer and polymer chemical structures and UV−vis studies.** Chemical structure of (**a**) mono-(S)-1 and mono-(S)-2. **b** Chemical structure of poly-(S)-1 and poly-(S)-2. **c** Comparison of experimental and theoretical UV−vis spectra of mono-(S)-1 and mono-(S)-2 in DCM and DMF. [mono-(R)-1] = 0.3 mg/mL, [mono-(R)-2] = 0.3 mg/mL. (Abs, absorbance).

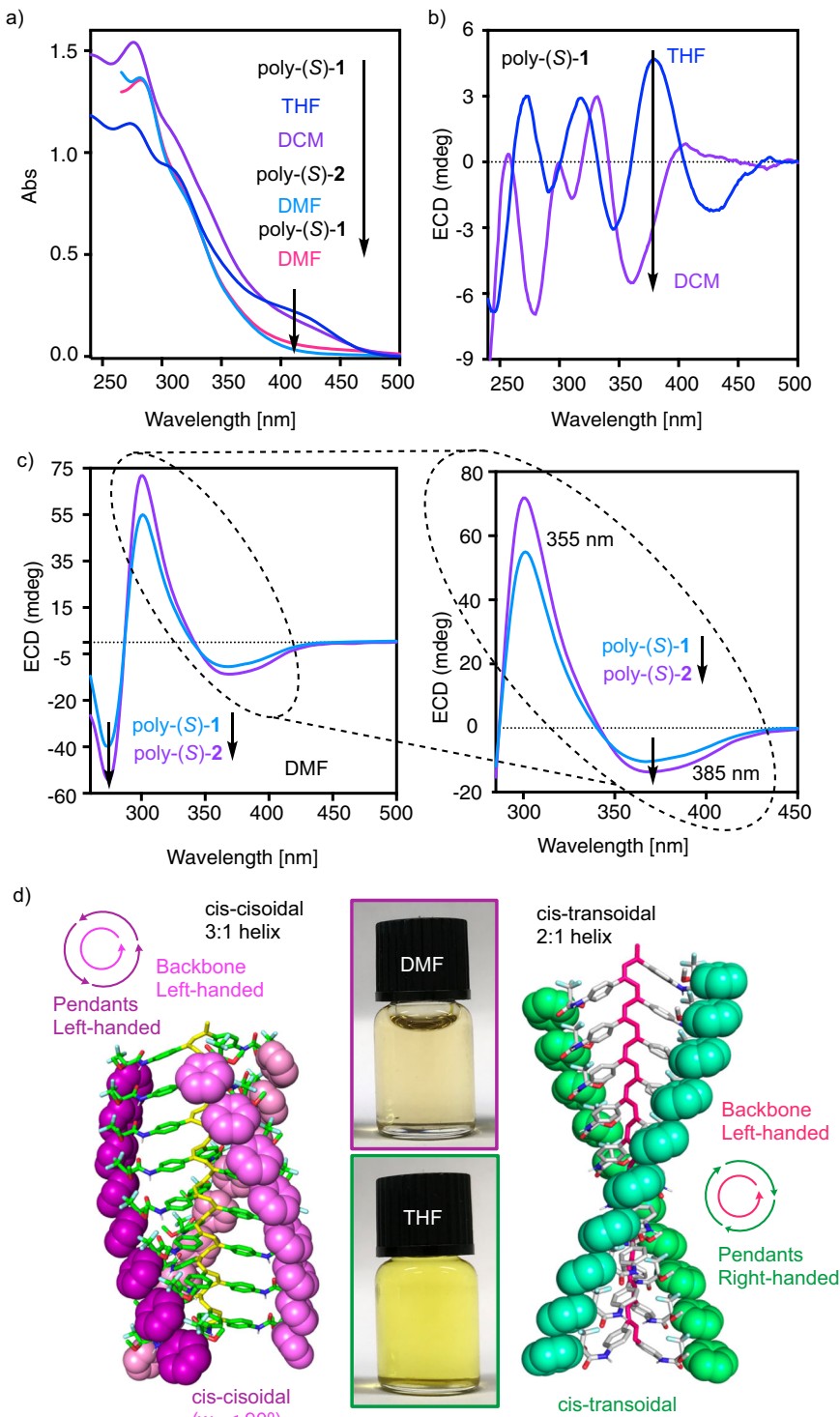

**Fig. 2 | ECD studies of poly-(S)-1 in different solvents and main scaffolds adopted by poly(phenylacetylene)s.** a) UV-vis spectra of poly-(S)-1 in THF, DCM and DMF and poly-(S)-2 in DMF. **b** ECD spectra of poly-(S)-1 in THF and DCM. **c** ECD spectra of poly-(S)-1 and poly-(S)-2 in DMF. **d** Schematic representation of the helical structure of PPA in a *cis-cisoidal* and a *cis-transoidal* conformation. [poly-(R)−**1**] = 0.6 mg/mL, [poly-(R)−**2**] = 0.6 mg/mL. (Abs, absorbance; ECD, electronic circular dichroism).

one. In the case of poly-(S)-**2**, similar results are obtained in DMF, whereas the experiments in THF or DCM were not possible due to insufficient solubility. The compression of the polyene backbone is accompanied by a color change from yellow to translucent due to the lack of alternating double bond conjugation of the polyene backbone (Fig. 2a). Note that the lack of emission, i.e. MMLCT luminescence,

both in DMF and DCM (see Supplementary Fig. 14) indicates the absence of short Pt·Pt interactions in the polymers[68–71].

ECD studies of poly-(S)-**1** and poly-(S)-**2** in these solvents show again an interesting effect. Thus, poly-(S)-**1** shows a complex ECD spectra in THF or DCM, with multiple bands in the 250–350 nm region (Fig. 2b). On the contrary, in polar solvents such as DMF, the ECD

spectrum is dominated by a strong bisignate in the 250−350 nm region (Fig. 2c). Careful inspection of the ECD spectrum of poly-(S)-1 in DMF reveals that the first negative Cotton band shows two minima, one centered at 385 nm and the other at 355 nm, which corresponds to the polyene backbone and the core of the Pt(II) complex, respectively (see above). Moreover, a strong positive Cotton band is obtained at 302 nm, and a negative one at 280 nm. More precisely, the band at ca. 302 nm corresponds to the *para*-ethynylpyridyl moiety, while the band at 280 nm is attributed to the *para*-pyridylbenzamide of (L)-valine-methyl ester, as previously discussed in the monomer calculations (Fig. 1c). The complexity of the ECD spectrum obtained for poly-(S)-1 in different solvents can be explained by combining the classical ECD spectrum of a poly(phenylacetylene) (PPA), which presents three alternating Cotton bands, in this case −/+/−, with the ECD trace obtained by a chiral supramolecular orientation of the metal containing spacers within the helical scaffold. Thus, while in DMF, poly-(S)-1 adopts a compressed *cis-cisoidal* structure ($\omega_1 < 90°$), in DCM or THF an elongated *cis-transoidal* structure ($\omega_1 > 90°$) is adopted by the polyene backbone. As a result, two different architectures with different structural properties are obtained for poly-(S)-1. The *cis-cisoidal* helix in PPAs is made by two coaxial helices that rotate in the same direction ($P_{int}/P_{ext}$ or $M_{int}/M_{ext}$), while the *cis-transoidal* helix rotates in the opposite direction ($P_{int}/M_{ext}$ or $M_{int}/P_{ext}$). Moreover, in the *cisoidal* scaffold, the helix contains three residues per turn, generating a helix with three crests at the periphery. For the *transoidal* scaffold, the helix possesses two residues per turn, describing a helix with two crests (Fig. 2d).

Interestingly, looking at the arrangement of the metal-containing spacers on the different crests of the *cis-cisoidal* and *cis-transoidal* helices, it is possible to create a unique helical arrangement of these achiral spacers within the helix crest whose sense depends on the helical scaffold. Accordingly, in *cis-cisoidal* PAs, the orientation of the achiral spacers within the helix crest exhibits the same sense as the internal and external helices. As a result, a multi-helical structure with five structural motifs is generated−internal helix, external helix and three helices described by the arrangement of the metal containing spacers at the helix crest.

From the literature, it is known that PPAs with a *P* orientation of the polyene backbone show an alternating ECD spectrum with three Cotton bands +/−/+, while those PPAs with an *M* orientation of the polyene backbone show the opposite orientation of the ECD bands (−/+/−). To determine how the axially chiral orientation of the metal containing monomer affects the spectrum, theoretical ECD studies were carried out in a supramolecular helical arrangement of an asymmetric bispyridyldichlorido Pt(II) complex. To this end, the geometry of supramolecular trimers arranged as left-handed and right-handed helices was optimized using DFT together with the rCAM-B3LYP functional and the LANL2DZ basis set[73–80]. In this case, the wB97XD functional was not employed considering the resemblance between the theoretical UV-vis obtained for the theoretical studies of (S)-1, and the computation time needed for the wB97XD functional.

Theoretical ECD calculations for both trimers (TD-DFT, rCAM-B3LYP/LANL2DZ) reveal a −/+ bisignate for the *M* helix and a +/− bisignate for the *P* helix. Interestingly, these two bands correspond to the core of the bispyridyldichlorido Pt(II) complex (350 nm) and to the *para*-ethynylpyridyl moiety (302 nm) (Fig. 3a).

Considering this result, if the polymer adopts a *cis-transoidal* scaffold with a preferred *P* orientation of the polyene backbone, the external helix and the axial array described by the metal-containing spacer will rotate in opposite direction (*M* helix). Thus, while the polyene produces a +/−/+ ECD trace, the axial array of the spacers promotes a −/+ ECD with an opposite orientation to that described by the polyene. As a result, depending on the intensity and wavelength absorbance of the first and second ECD bands produced by the polyene backbone and the axial array of the metal-containing spacer,

complex ECD spectra are obtained with multiple positive and negative ECD bands as observed in poly-(S)-1 dissolved in DCM or THF (Fig. 2b).

On the contrary, in a *cis-cisoidal* helix with a *P* orientation of the internal and the external helices, and a *P* orientation of the asymmetric Pt(II) complexes used as spacers, the first and second cotton bands of the classical ECD spectrum (+/−/+) experience an additive effect of the (+/−) bands obtained for the axial array of the spacers.

As a result, an ECD spectrum with three alternating Cotton bands is obtained, where the second is very intense.

In the case of a polymer adopting a *cis-cisoidal* structure with a preferred *M* screw sense of the polyene backbone, the opposite −/+/− ECD trace must be obtained. This is in fact the ECD pattern that we observed for poly-(S)-1 dissolved in DMF: three alternating Cotton bands (−/+/−), showing the second and third band an abnormally large intensity, confirming the additive effect by ECD theoretical calculations (TD-DFT(CAM-B3LYP)/LANL2DZ). Analogous spectra were obtained for poly-(S)-2 dissolved in DMF (Fig. 3b and Supplementary Information).

VT-ECD studies of poly-(S)-1 in THF and DMF corroborate these structures, showing for poly-(S)-1 in DMF a large thermal stability (above 80 °C) promoted by the three axial arrays of the asymmetric Pt(II) complexes used as spacers. Thermal studies for poly-(S)-1 in THF disclose a lower thermal stability due to the presence of just two axial arrays of the asymmetric bispyridyldichlorido Pt(II) complexes used as spacers within the helical scaffold.

## Discussion

In summary, we have achieved through two different examples−poly-(S)-1 and poly-(S)-2−unique multi-helical materials built from a supramolecular helix, made from a bispyridyldichlorido Pt(II) complex and a covalent helical polymer (PA). When poly-(S)-1 is dissolved in solvents of moderate polarity, e.g. THF, the polyene backbone adopts a *cis-transoidal* helical scaffold with four axial motifs: the internal helix (helix 1) described by the polyene backbone, the external helix (helix 2) described by the pendants and two helices described by the bispyridyldichlorido Pt$^{II}$ complexes used as spacers (helices 3 and 4). These four axial motifs are interconnected describing either a $M_1/P_2/P_3/P_4$ or $P_1/M_2/M_3/M_4$ multi-axial chiral system. Interestingly, an additional axial motif emerges when the polymer is dissolved in polar solvents such as DMF. In such case, poly-(S)-1 adopts a *cis-cisoidal* scaffold that shows five axial motifs: the internal helix (helix 1), the external helix (helix 2), and axial motifs described by the bispyridyldichlorido Pt$^{II}$ complexes used as spacers (helices 3 and 4). These structural motifs are interconnected describing either a $P_1/P_2/P_3/P_4/P_5$ or $M_1/M_2/M_3/M_4/M_5$ multi-axial chiral material. These results indicate that rational molecular design can be exploited to stabilize supramolecular helices in covalent polymers, which paves the way for the design of complex multi-helical materials.

## Methods

*CD measurements* were performed on a Jasco-720 and VT-CD measurements on a Jasco-1500 with a 1 mm quartz cuvette. The amount of polymer used for CD and VT-CD measurements was 0.6 mg/mL.

*UV spectra* were registered on a Jasco V-750 with a 1 mm quartz cuvette. The amount of polymer used for UV measurements was 0.6 mg/mL.

*Optical rotation* was measured on a Jasco-P2000.

*Mass spectra* were obtained on an HP 5988A spectrometer (Hewlett-Packard, Palo Alto, CA, USA).

*Elemental analyses* were recorded using a Foss-Heraeus CHNO-Rapid analyzer.

*NMR experiments* were measured on a Varian 300 (1H: 300 MHz) or a Bruker Avance II 400 (1H: 400 MHz; 13 C: 101 MHz; 19 F: 376.4 MHz). Solid NMR experiments were measured on a Bruker NEO 750 using a 1.3 mm rotor.

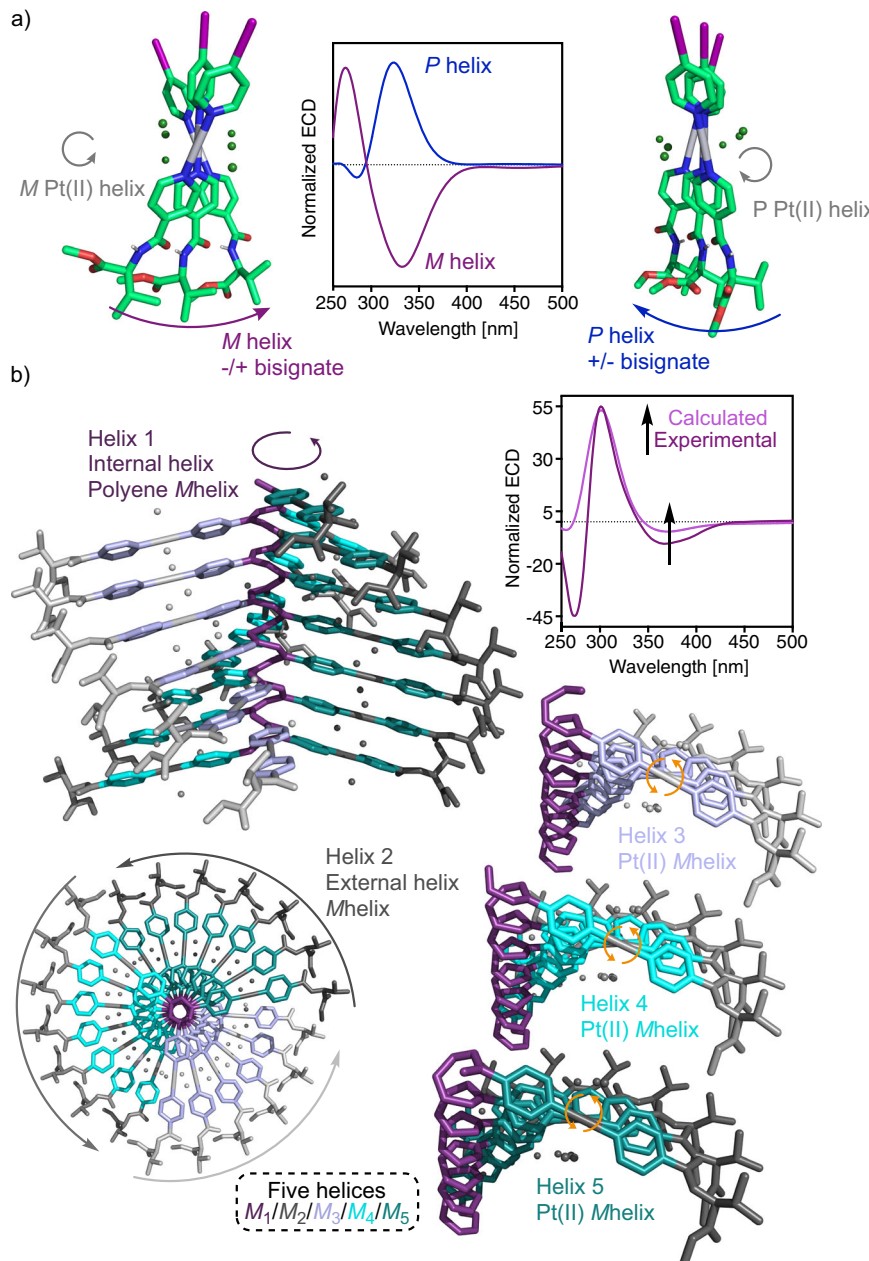

**Fig. 3 | Theoretical ECD studies and approximated 3D structure for poly-(S)-1.**
**a** Representation of the calculated models for the assembly of three units of mono-(S)-1 in a left-handed and a right-handed helix with the calculated ECD spectra.

**b** Illustration and ECD of the 3D model of a *c-c* helix of poly-(S)-1 resulting in a multihelical material composed of 5 helices. [poly-(R)-1] = 0.6 mg/mL. (ECD, electronic circular dichroism).

*Raman spectra* were performed on a Renishaw confocal Raman spectrometer (Invia Reflex model), equipped with two lasers (diode laser 785 nm and Ar laser 514 nm).

*DSC traces* were obtained on a DSC Q200 Tzero Technology (TA Instruments, New Castle, UK), equipped with a refrigerated cooling system RCS90 (TA Instruments, New Castle, UK), using a Tzero low-mass aluminum pan.

*TGA traces* were obtained on a TGA Q5000 (TA Instruments, New Castle, UK) using a platinum pan.

*Photoluminescence spectra* were obtained on a Horiba Fluoromax-Plus-C fluorimeter using a 10 mm quartz cuvette.

*IR spectra* were obtained on a JASCO-FTIR-6800. IR spectra in solution were recorded using a CaF2 cell with a path length of 0.1 mm.

*Computational studies*: 3D structures were done with the software package Spartan´18. The asymmetric bispyridyldichlorido Pt(II) complex was built by using the data obtained from another bispyridyldichlorido Pt(II) complex (CCDC deposit number 2004415). Time-dependent density functional theory (TD-DFT) together with either rCAM-B3LYP and wB97XD density functional and LANL2DZ basis set [TD-DFT(wB97XD)/LANL2DZ, TD-DFT(rCAM-B3LYP)/LANL2DZ)] were carried out with the software package Gaussian 16, rev C.01.

### Synthesis of ligands

*Methyl isonicotinoyl-L-valinate [Ligand 1]*: Isonicotinic acid (441 mg, 1.20 equiv.), 1-ethyl-3-(3-dimethylaminopropyl)carbodiimide (EDC, 686 mg, 1.20 equiv.), 1-hydroxybenzotriazole (HOBt, 484 mg, 1.20 equiv.) and 4-dimethylaminopyridine (DMAP, 437 mg, 1.20 equiv.) were dissolved in 30 mL of dry CH2Cl2. After 15 min, time needed to activate the acid, L-valine methyl ester (500 mg, 1.00 equiv.) was added and the mixture was stirred overnight. The organic layer was washed

three times with HCl 1 M and a saturated solution of NaHCO$_3$. The combined organic layers were dried over anhydrous Na$_2$SO$_4$, filtered and evaporated at reduced pressure. The crude product was chromatographed on silica gel with pentane/ethyl acetate (70:30) as eluent (388 mg, 55 % of yield).

*Methyl isonicotinoyl-L-alaninate [Ligand 2]:* Isonicotinic acid (529 mg, 1.20 equiv.), 1-ethyl-3-(3-dimethylaminopropyl)carbodiimide (EDC, 824 mg, 1.20 equiv.), 1-hydroxybenzotriazole (HOBt, 581 mg, 1.20 equiv.) and 4-dimethylaminopyridine (DMAP, 525 mg, 1.20 equiv.) were dissolved in 30 mL of dry CH$_2$Cl$_2$. After 15 min, time needed to activate the acid, L-alanine methyl ester (500 mg, 1.00 equiv.) was added and the mixture was stirred overnight. The organic layer was washed three times with HCl 1 M and a saturated solution of NaHCO$_3$. The combined organic layers were dried over anhydrous Na$_2$SO$_4$, filtered and evaporated at reduced pressure. The crude product was chromatographed on silica gel with pentane/ethyl acetate (70:30) as eluent (395 mg, 53 % of yield).

*4-((trimethylsilyl) ethynyl)pyridine [Ligand 3]:* 4-Iodopyridine (500 mg, 1.00 equiv.), bis(triphenylphosphine)palladium (II) dichloride (Pd(PPh$_3$)$_2$Cl$_2$, 6.85 mg, 0.04 equiv.), triphenylphosphine (10.2 mg, 0.016 equiv.) and copper iodide (CuI, 11.1 mg, 0.024 equiv.) were dissolved in dry THF (20 mL). Next, triethylamine (Et$_3$N, 10 mL) and ethynyltrimethylsilane (359 mg, 1.5 equiv.) were added and the mixture was stirred for two hours. After removing the solvent, the crude product was chromatographed on silica gel with pentane/ethyl acetate (80:20) as eluent obtaining, after solvent removal, an oil (368 g, 86 % of yield). The synthetic procedure was previously reported in supplementary reference 1.

### General protocol for complexation and deprotection of monomers

Ligand 3 (1.7 equiv.), trans-PtCl$_2$(PhCN)$_2$ (1.35 equiv.) and the corresponding ligand 1 or 2 (1 equiv.) were placed in a pressure tube and subjected to five vacuum/Ar cycles. Subsequently, dry and degassed toluene was added and the mixture was heated to 70 °C and stirred at this temperature for one week. After evaporating the solvent, the crude was chromatographed on silica gel with pentane/ethyl acetate (80:20) as eluent.

The corresponding Pt(II) complexes with the protected alkyne were dissolved in a mixture of THF:MeOH (3:1). Next K$_2$CO$_3$ was added (1.20 equiv.) and the reaction was stirred at r.t. for 1 h. The crude was chromatographed on silica gel with pentane/ethyl acetate (60:40) as eluent.

### Synthesis of polymers

The reaction flask (sealed ampoule) was dried under vacuum and argon flushed for three times before monomer was added as a solid. Dry THF was added with a syringe. Next, a solution of rhodium norbornadiene chloride dimer, [Rh(nbd)Cl]$_2$, and Et$_3$N in dry THF was added to the reaction under stirring at 30 °C. After 6 h, the resulting polymer was diluted in CH$_2$Cl$_2$ and precipitated in methanol and centrifuged twice. Next, the polymer was reprecipitated in hexane and centrifuged again.

### Data availability

All data is available from the corresponding authors upon request.

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

## Acknowledgements

The authors thank Servicio de Microscopía Electrónica (RIAIDT, USC). Financial support from AEI (PID2019–109733GB-I00, E.Q. and F.F.), Xunta de Galicia (ED431C 2018/30, ED431C 2021/40, Centro Singular de Investigación de Galicia acreditación 2019–2022, ED431G 2019/03; E.Q. and F.F.) and the European Regional Development Fund (ERDF) is gratefully acknowledged. F.R.T. thanks Xunta de Galicia for a predoctoral contract.

## Author contributions

F.F. and G.F. conceived the project. F.R.-T. synthesized all materials and performed all the experimental studies as well as the theoretical calculations. F.R.-T., E.Q., F. F. and G.F. discussed the results and wrote the manuscript with input from all authors.

## Funding

## Competing interests

The authors declare no competing interests.
