## [Peer review file · Nature Communications]

REVIEWER COMMENTS

Reviewer #1 (Remarks to the Author):

Freire et al. reported novel multi-chiral polymers that contain both noncovalent supramolecular chiral structures and covalent chiral helical backbones. In this system, the helical structure described by the poly(acetylene) (PA) backbone (cis-cisoidal, cis-transoidal) guides the pendant groups in a fashion where a tilting degree emerges between a pendant and the adjacent ones at the helical crest, in this case, the bispyridyldichloride Pt(II) complex. It should be pointed out that in these results, the helical conformation of poly(acetylene) (PA) backbone can drive the side-chain supramolecules to form helical structures even in solution. Adjusting the conformation of the PA backbone (cis-cisoidal, cis-transoidal) by solvent, multi-helical polymers can be designed comprising four or five different helices—internal and external helix or multi-helical bispyridyldichloride Pt(II) complexes. The manuscript is clear to read and presents a good scholarly result overall. The results are interesting for a broad and general readership in polymer, chirality and supramolecular chemistry. It is a nice work and I think it will attract citations and interest. The following recommendations are for the author's consideration.

1. Regarding the "material" mentioned in the title, although this polymer exhibits interesting chiroptical activity, practical applications of this material should be stated in detail.
2. The bispyridyldichloride Pt(II) complex has poor solubility, it is difficult to confirm that this polymer is in a molecularly dissolved state. Whether other optically active conjugated molecules can be used as suitable candidates, for example azobenzene, perylene or BTA. What the basis?
3. The compression of the polyene backbone is accompanied by a color change from yellow to translucent due to the lack of alternating double bonds conjugation of the polyene backbone. The photo of Figure 2d shows an interesting color change, in this case, are optically active polymer particles formed?
4. By rational molecular design, it is possible to stabilize supramolecular helices in covalent polymers. In this case, the chiroptical properties of the polymer can also be regulated by external stimuli, such as metal ions, can these two polymers dynamically respond to metal ions with different valence states?
5. Chiral supramolecular array of conjugated groups in the spacer, not the flexible alkyl chain, was driven by the helical conformation of PA backbone. But for References [53-56], the supramolecular chiral arrays of azo groups may not originate from the covalent backbone helix. In these works, the polymethacrylates main-chains were prepared by radical polymerization, the tacticity of the polymer backbone is inherently not stereospecific. The involved polymer contains four important units: 1) the main-chain of methacrylate; 2) alkyl spacer (such as hexadecyl); 3) mesogens and 4) alkyl chiral tails. A methacrylate main-chain was chosen because of its flexibility in polymers. To preserve the flexibility of the main-chains, alkyl linkers were also inserted between the main-chain and the conjugated moieties. The discussed Azo helical stacks originated from liquid-crystalline (LC) chirality of polymer systems, such as chiral doping in achiral LC polymers. The induced supramolecular chirality arises from superstructural

chirality, i.e., helical structure via aggregation of side-chain Azo units, and is independent of the conformation of polymer main-chains.

6. Some papers are suggested to be cited: Supramolecular arrangements in side-chain LC polymers, J. Am. Chem. Soc. 2017, 139, 13218-13226; Macromolecules 2022, 55, 8556-8565.

Reviewer #2 (Remarks to the Author):

This manuscript describes unique multi-helical structures formation using Pt(II)-based metallo-supramolecular polymers.

The title and the abstract clearly shows the main research results.

Synthetic procedures and characterization data of the ligand and the Pt(II) complexes are perfectly presented in SI. However, as for the polymer synthesis, I think authors do not provide an evidence that the Pt(II) complex moieties exist without decomplexation after the polymerization reaction. The broad peaks in Figures S7 and S8 suggest the polymer formation, but I think it is difficult from the spectra to judge whether the decomplexation of the Pt(II) complex moieties occurred or not.

The analysis of ECD spectra and the helical structure estimation with the help of the calculation are fine, but I think the addition data to support the estimated structures shown in Figures 2 and 3 is necessary. Are there any other possible structures except for the proposed helical structures? For example, are there any inter-polymer metal-metal interactions among the polymers?

After all, the proposed helical system is interesting, but I think the data to support the hypothesis is not enough. For example, in order to investigate the origin of the helical changes deeply, authors should synthesize the different chain length (short length) of the polymers (or the oligomers) and compare the ECD spectra with those of the present long polymers.

Reviewer #3 (Remarks to the Author):

This manuscript reports the characterization of stereo structure of two polyphenylacetylene derivatives in which the main-chain structure was investigated from a view of cis-cisoidal (shrunk) and cis-transoidal (stretched) structures and also from a view of left- and right-handed helicity and the side-chain chirality was viewed from a view of left- and right-handed helicity. Chirality of helical polymers is generally recognized as a 'total' property that may be contributed by different aspects of chirality and such

aspects are not discussed separately. On the other hand, the authors present detailed analyses separating the contributions from the structural features said above to the CD spectra based utilizing DFT calculations (for rather small systems up to model trimers) as well as classical or traditional, empirical views. The authors are challenging this generally difficult issue of polymer helix in a rational way, and at least, as far as the polymers dealt with in this work are concerned, the interpretation of the CD spectra seems reasonable. I feel that the present work is worth being considered positively by the editor for publication.

One specific point: The functionals, rCAM-B3LYP and wB97XD, were used for different systems. If the authors could give very brief reasoning why they used the two different functionals for the relevant systems, the work would be even more convincing and informative.

Reference citation: there are papers about establishing polymer helix of other chemical structures such as polyacrylates, polyfluorenes, poly(fluorenevinylene)s, and so on by other types of theoretical analyses, and relevant ones should be considered for citation.

Reviewer 1:

Reviewer 1 states: - 1. Regarding the "material" mentioned in the title, although this polymer exhibits interesting chiroptical activity, practical applications of this material should be stated in detail.

Our answer: Axial chirality plays an important role in the application of helical polymers in different fields such as chiral HPLC (helical polymer used as chiral stationary phase), asymmetric catalysis or chiral recognition among others. Here, we present an interesting family of helical polymers, which possesses up to five different chiral axial motifs within the helical scaffold and, therefore, considering the structure-function relationship, we strongly believe that the efficiency of these type of materials can be improved by increasing the axial motifs within the helix. This is our next goal in this research line.

Reviewer 1 states: - The bispyridyldichloride Pt(II) complex has poor solubility, it is difficult to confirm that this polymer is in a molecularly dissolved state. Whether other optically active conjugated molecules can be used as suitable candidates, for example azobenzene, perylene or BTA. What the basis?

Our answer: The bispyridyldichlorido Pt(II) complex has an excellent solubility in the monomeric state in typical organic solvents. Once this monomer is polymerized, its solubility is highly dependent on the solvent used to dissolve it. Thus, poly-1 is perfectly soluble in solvents such as DCM, THF or DMF. Linear Dichroism (LD) has been measured to demonstrate that the ECD signal is due to the presence of a helical structure and not due to the presence of aggregates (See Figure S16).

In this case, we are very interested in using bispyridyldichlorido Pt(II) complexes as spacers between the chiral center and the polymerizable alkyne group due to its linear shape, and the presence of metal ions, which allow us to create the desired helical metal-containing polymer backbones, a family of polymers that has remained scarcely studied thus far.

We find the referee's suggestion of using azobenzene, BTAs or perylenes very interesting. We would like to highlight that our current submission is a proof-of-concept work, which we plan to extend to other molecular platforms in the near future. Thus, the use of multiple chromophores is certainly an interesting approach for further submissions. However, these systems may exhibit different properties, including planarity, which may be too bulky to be used as spacers and keep the dynamic behavior of the polymer. Moreover, when one is dealing with dynamic helical polymers, the use of different building blocks changes the overall properties due to the triggering of different cooperative effects, which can produce changes in the structure, stimuli-responsiveness and helical induction effects. Thus, different families of compounds must be analyzed independently; and check how this functional group and its position in the monomer structure affects the dynamic behavior of the PPA.

Reviewer 1 states: - The compression of the polyene backbone is accompanied by a color change from yellow to translucent due to the lack of alternating double bonds conjugation of the polyene backbone. The photo of Figure 2d shows an interesting color change, in this case, are optically active polymer particles formed?

Our answer: The polymer was perfectly soluble in both solvents. The newly recorded linear dichroism (LD) experiments are silent (see Figure S16), which indicates that the ECD trace is due to the helical structure adopted by the polymer and not by the presence of large aggregates.

Reviewer 1 states: - By rational molecular design, it is possible to stabilize supramolecular helices in covalent polymers. In this case, the chiroptical properties of the polymer can also be regulated by external stimuli, such as metal ions, can these two polymers dynamically respond to metal ions with different valence states?

Our answer: As suggested by the referee, different metal ions were added to a solution of poly-(S)-1, but no structural effects such as helix enhancement, helix inversion or differences in the polymer elongation have been observed. The ECD signal was reduced, probably due to a dilution effect after adding a solution of the metal ion.

In this case, we just found that poly-(S)-1 responds to solvents obtaining different ECD traces in THF, DCM and DMF.

Figure. ECD and UV-vis spectra of poly-(S)-1 in (a) DCM and (b) THF ($c = 0.6$ mg/mL) before and after the addition of different metal perchlorate salts [$M(\text{ClO}_4)_n = 10$ mg/mL MeOH].

Reviewer 1 states: Chiral supramolecular array of conjugated groups in the spacer, not the flexible alkyl chain, was driven by the helical conformation of PA backbone. But for References [53-56], the supramolecular chiral arrays of azo groups may not originate from the covalent backbone helix. In these works, the polymethacrylates main-chains were prepared by radical polymerization, the tacticity of the polymer backbone is inherently not stereospecific. The

involved polymer contains four important units: 1) the main-chain of methacrylate; 2) alkyl spacer (such as hexadecyl); 3) mesogens and 4) alkyl chiral tails. A methacrylate main-chain was chosen because of its flexibility in polymers. To preserve the flexibility of the main-chains, alkyl linkers were also inserted between the main-chain and the conjugated moieties. The discussed Azo helical stacks originated from liquid-crystalline (LC) chirality of polymer systems, such as chiral doping in achiral LC polymers. The induced supramolecular chirality arises from superstructural chirality, i.e., helical structure via aggregation of side-chain Azo units, and is independent of the conformation of polymer main-chains.

Some papers are suggested to be cited: Supramolecular arrangements in side-chain LC polymers, J. Am. Chem. Soc. 2017, 139, 13218-13226; Macromolecules 2022, 55, 8556-8565

Our answer: We agree with the reviewer, and this paragraph has been modified according to his/her suggestions. References 57-58 have been renumbered as references 53 and 54. Macromolecules 2022, 55, 8556-8565 has been added as reference 55.

The new references 56-59 (before 53-56) have been added in addition to the new reference, J. Am. Chem. Soc. 2017, 139, 13218-13226 (reference 60), at the end of the following paragraph inserted on page 2, line 5:

“Interestingly, in literature there are also other examples where a chiral supramolecular helical array of the pendant groups is produced, although the polymer main chain is not folded into a preferred screw sense helix.⁵⁵⁻⁶⁰”

References 58-70 were renumbered as references 60-72.

Reviewer 2:

Reviewer 2 states: Synthetic procedures and characterization data of the ligand and the Pt(II) complexes are perfectly presented in SI. However, as for the polymer synthesis, I think authors do not provide an evidence that the Pt(II) complex moieties exist without decomplexation after the polymerization reaction. The board peaks in Figures S7 and S8 suggest the polymer formation, but I think it is difficult from the spectra to judge whether the decomplexation of the Pt(II) complex moieties occurred or not.

Our answer: The chiral group is connected to the polyene backbone through the Pt(II) complex. Decomplexation of the metal ion would imply that the polymer lacks the chiral group and therefore it cannot adopt a screw sense excess (ECD active). After polymerization reaction, the polymer is purified by precipitation in methanol and reprecipitation in hexane. In those solvents, the unreacted monomer is removed from the solution. Therefore, in experiments such as NMR or Raman, the peaks associated with the chiral ligand and the metal correspond to metal attached to the polymer. In addition, NMR studies show the presence of the chiral amino acids in the polymer, which are needed to induce a helical sense into the PPA (Figure S7b). The integration of the proton peaks, see below, yields a 1:1 correlation between the protons of the chiral amino acid and the polymer backbone, which indicates that is still coordinated after polymerization. In the Raman spectra of the polymer (Figure S9), the band at ca. 330 cm⁻¹ corresponds to the Pt-Cl vibration, which further corroborates that the metal is present in the final polymer. Finally, if decomplexation had occurred, emission corresponding to free pyridine ligands would have been observed, which is not the case (Figure S14). In short, we conclude based on various experimental methods that decomplexation does not occur.

^1H NMR spectra of poly-(S)-**1** in CDCl_3 (400 MHz).

Reviewer 2 states: The analysis of ECD spectra and the helical structure estimation with the help of the calculation are fine, but I think the addition data to support the estimated structures shown in Figures 2 and 3 is necessary. Are there any other possible structures except for the proposed helical structures? For example, are there any inter-polymer metal-metal interactions among the polymers?

Our answer: By ECD and UV-vis studies we observed changes in the polymer elongation due to a bathochromic/hypsochromic shift of the polyene backbone in different solvents, associated to changes in the conjugation of the double bonds that form the polyene skeleton. CD studies do not show variations in the helical sense of the polymer. Linear dichroism studies (Figure S16) do not show any signal, which indicate that the ECD trace is due to the helical polymer.

Regarding the existence of metal-metal interactions among the polymers, it is known that bispyridyldichlorido Pt(II) complexes are relatively bulky due to the out-of-plane arrangement of the Cl ligands with respect to the pyridine rings, preventing close Pt-Pt contacts upon self-assembly (see for instance Chem. Sci., 12, 12248-12265 (2021)). These materials are non-emissive. Only if multiple hydrogen bonding groups with sufficient conformational freedom (such as bisamides) are included in the molecular design, $^3\text{MMLCT}$ luminescence arises as a result of Pt-Pt interactions in the excited state. In fact, to the best of our knowledge, there is only one report on MMLCT-luminescent assemblies of bispyridyldichlorido Pt(II) complexes, which manifests by the appearance of a broad, featureless emission band between ca. 500 and 750 nm (Angew. Chem. Int. Ed., 61, e202208436 (2022)). As our current system (poly-(S)-**1**) is non-emissive in the investigated solvents (see photoluminescence studies in Figure S15), this rules out the presence of short Pt-Pt interactions in the polymer. We have added a new sentence to the main text to clarify this point.

Reviewer 2 states: in order to investigate the origin of the helical changes deeply, authors should synthesize the different chain length (short length) of the polymers (or the oligomers) and compare the ECD spectra with those of the present long polymers

Our answer: To address this comment, a new section has been included in the S.I.:

Polymerization degree effects (S2o)

To determine the role of the polymerization degree in the chiroptical properties of poly-(S)-1, an oligomer of poly-(S)-1 was prepared according to Prof. Maeda's protocol. (ref. S6: T. Taniguchi, T. Yoshida, K. Echizen, K. Takayama, T. Nishimura, K. Maeda, Angew. Chem. Int. Ed. 2020, 59, 8670-3680)

GPC studies show the presence of two different peaks at 19 and 26 min, which correspond to a polymer (DP = 247) and an oligomer (DP = 24) of poly-(S)-1, respectively. ECD studies show in both cases the same ECD trace, although this is more intense for the polymer (g_{abs} (poly-(S)-1) = 1.05×10^{-3} ; oligomer-(S)-1 = 6.87×10^{-4}). This fact indicates that in large polymers, the relationship between monomer repeating units at internal positions vs. the helix edge is larger than in the oligomer. As a result, there are more bispyridyldichlorido platinum(II) intrapendant supramolecular interactions, that result in a large ECD spectrum

a)

	Mn	Mw	Mn/Mw
poly-(S)-1 (247mer)	9,51309E+04	1,10973E+05	1,17
poly-(S)-1 (24mer)	1,4900E+04	1,55E+04	1,04

b)

Figure S2o. a) GPC traces and (b) spectroscopic studies (ECD, UV-vis) of an oligomeric and polymeric version of poly-(S)-1. [poly-(S)-1] = 0.6 mg/mL DMF.

Reviewer 3:

Reviewer 3 states: The functionals, rCAM-B3LYP and wB97XD, were used for different systems. If the authors could give very brief reasoning why they used the two different functionals for the relevant systems, the work would be even more convincing and informative.

Our answer: During the computational studies of the monomer, both functionals rCAM-B3LYP and wB97XD were used to reproduce the experimental UV-vis spectra. In this case, both

functionals show a good match with the experimental one, although the best fit was obtained with wB97XD (Figure S18).

Figure S18. Experimental and theoretical UV-vis spectra of mono-(S)-1

The corresponding paragraph in the main text (page 3, line 7) was modified as follows:

“UV-vis theoretical calculations for mono-(S)-1 (time-dependent density functional theory (TD-DFT) together with either rCAM-B3LYP and wB97XD density functional and LANL2DZ basis set [TD-DFT(wB97XD)/ LANL2DZ], TD-DFT(rCAM-B3LYP)/LANL2DZ], were performed on the molecular structure obtained after DFT optimization (see S.I. for details). Both theoretical calculations show a good match between the experimental and the calculated ECD trace, being slightly better when the functional wB97XD is used. From the TD-DFT (wB97XD)/ LANL2DZ theoretical studies, we identified which fragments of mono-(S)-1 are responsible for these absorption bands.”

The good match obtained from both theoretical calculations [TD-DFT(wB97XD)/ LANL2DZ], TD-DFT(rCAM-B3LYP)/ LANL2DZ], drove us to use the rCAM-B3LYP functional to analyze the axially chiral oligomeric supramolecular array. This selection was done to optimize the computational time needed during these theoretical studies.

A new sentence was added to the main text (page 7, line 14):

“In this case, the wB97XD functional was not employed considering the resemblance between the theoretical UV-vis obtained for the theoretical studies of (S)-1, and computation time needed for the wB97XD functional.”

Reviewer 3 states: there are papers about establishing polymer helix of other chemical structures such as polyacrylates, polyfluorenes, poly(fluorenevinylene)s, and so on by other types of theoretical analyses, and relevant ones should be considered for citation.

Our answer: References 73-79 were added to the main text.

REVIEWERS' COMMENTS

Reviewer #1 (Remarks to the Author):

The revised manuscript is detailed and convincing. I believe in the correctness of the understanding of the results they show in the paper. Therefore, I think the current manuscript deserves to be published.

Reviewer #2 (Remarks to the Author):

Authors have answered my comments clearly and added a data on the effect of the molecular weight to the helical structures according to my request. So, I recommend the revised manuscript is published in this journal.

Reviewer #3 (Remarks to the Author):

The manuscript seems to have been revised sufficiently except for the changes in the reference section.

As for citation regarding helical structural establishment by computation, the added papers, refs. 76 and 77, seem not necessarily relevant. Instead of 76 and 77, the following papers should be considered for citation:

Angew. Chem. Int. Ed. Engl. 2022, 61 (43), e202210556.

Angew. Chem. 2015, 127 (9), 2726-2730.

Journal of the American Chemical Society 2013, 135 (15), 5509-5512.

POINT-BY-POINT RESPONSE TO THE REFEREES' COMMENTS

Reviewer 1:

The revised manuscript is detailed and convincing. I believe in the correctness of the understanding of the results they show in the paper. Therefore, I think the current manuscript deserves to be published.

Our answer: We thank the referee for supporting final publication of our work.

Reviewer 2:

Authors have answered my comments clearly and added a data on the effect of the molecular weight to the helical structures according to my request. So, I recommend the revised manuscript is published in this journal.

Our answer: We thank the referee for supporting final publication of our work.

Reviewer 3:

The manuscript seems to have been revised sufficiently except for the changes in the reference section.

As for citation regarding helical structural establishment by computation, the added papers, refs. 76 and 77, seem not necessarily relevant. Instead of 76 and 77, the following papers should be considered for citation:

Angew. Chem. Int. Ed. Engl. 2022, 61 (43), e202210556.

Angew. Chem. 2015, 127 (9), 2726-2730.

Journal of the American Chemical Society 2013, 135 (15), 5509-5512.

Our answer: We thank the referee for supporting final publication of our work. References 76 and 77 have been replaced by the suggested ones.